# Identifying the Pathological Domain of Alpha- Synuclein as a Therapeutic for Parkinson’s Disease

**DOI:** 10.3390/ijms20092338

**Published:** 2019-05-11

**Authors:** Ning Shen, Ge Song, Haiqiang Yang, Xiaoyang Lin, Breanna Brown, Yuzhu Hong, Jianfeng Cai, Chuanhai Cao

**Affiliations:** 1Department of Chemistry, University of South Florida, Tampa, FL 33612, USA; ningshen@mail.usf.edu (N.S.); haiqiangyang@mail.usf.edu (H.Y.); jianfengcai@usf.edu (J.C.); 2Department of Surgery of Traditional Chinese Medicine, Tianjin University of Traditional Chinese Medicine, Tianjin 300193, China; songgegs@gmail.com; 3Department of Pharmaceutical Sciences, College of Pharmacy, University of South Florida, Tampa, FL 33612, USA; xlin@health.usf.edu (X.L.); breannabrown@mail.usf.edu (B.B.); yuzhu1@mail.usf.edu (Y.H.); 4Department of Neurology, College of Medicine, University of South Florida, Tampa, FL 33612, USA

**Keywords:** Alpha synuclein, Parkinson’s disease, epitope, antibody, antigenicity, immunotherapy

## Abstract

Alpha-synuclein is considered the major pathological protein associated with Parkinson’s disease, but there is still no effective immunotherapy which targets alpha-synuclein. In order to create a safer and more effective therapy against PD, we are targeting an epitope of alpha-synuclein rather than full-length alpha-synuclein. We have selected several antigenic domains (B-cell epitope) through antigenicity prediction, and also made several recombinant protein fragments from alpha-synuclein upon antigenicity prediction in an E. coli system. We then tested the function of each of the peptides and recombinant fragments in aggregation, their toxicity and antigenicity. We have discovered that the full-length recombinant (aa1–140) can aggregate into oligomers or even fibrils, and fragment aa15–65 can promote the aggregation of aa1–140. It is worth noting that it not only promotes whole protein aggregation, but also self-aggregates as seen by western blotting and silver staining assays. We have tested all candidates on primary neurons for their toxicity and discovered that aa15–65 is the most toxic domain compared to all other fragments. The antibody targeting this domain also showed both anti-aggregation activity and some therapeutic effect. Therefore, we believe that we have identified the most potent therapeutic domain of alpha synuclein as a therapeutic target.

## 1. Introduction

Parkinson’s disease (PD) is the second most common neurodegenerative disorder [1]. The history of PD’s discovery dates back to 1817, and it was first described as a “shaking palsy” by Dr. James Parkinson [2]. A particularly strong link between PD and α-synuclein has been identified with α-synuclein being the pathological hallmark of PD. Lewy bodies (LBs) and Lewy neurites (LNs) were attributed to the aggregated forms of α-synuclein accumulates [3]. A-synuclein is a small protein (140 amino acids) mainly located in the presynaptic terminals of neurons and is thought to be critical for synaptic function and plasticity, as well as vesicular packaging and trafficking [4,5,6] by regulating vesicle number [7] and adjusting their assembly [8,9,10]. SNCA gene mutations, which lead to α-synuclein protein misfolding, were the first demonstrated genetic cause of familial Parkinsonism with Lewy pathology [11]. Also, under some pathogenic conditions, α-synuclein tends to form oligomers, which have high toxicity and induce cell death [12]. Articles discussing Lewy bodies revealed that the clinical progression of PD correlates with α-synuclein progressively spreading and aggregating [13,14,15]. It is strongly suggested that the spread of aggregated α-synuclein leads to the progression of PD. Therefore, stopping the spread of aggregated α-synuclein might be the best approach, or even the solution, for treating PD.

Significant advances have been made in the fields of PD therapy, including pharmacologic agents, genetic engineering, and cell replacement therapeutic approaches [16]. Although remarkable progress has been achieved, these treatments are only able to reduce the severity of PD, often with serious side effects. Over the course of human history fighting diseases, the most successful innovation has been vaccine development and application. Vaccines are a safe, economical, and the ultimate approach for eradicating diseases. The smallpox vaccine has saved the world, and the hepatitis B (HBV) vaccine has relieved a large human financial burden across the world. Vaccine progress against disoriented proteins, such as the prion [17,18,19,20,21,22] and amyloid beta [23,24,25], inspired scientists to develop active and passive immunotherapies for PD. Masliah et al. indicated that immunization of PD transgenic (PD/Tg) mice with full-length human α-synuclein could reduce misfolded α-synuclein accumulates in neuronal cell bodies and synapses, and also reduced neurodegeneration in a human α-synuclein transgenic PD mouse model [26]. Since the full-length α-synuclein protein is a normal product found in the body, therapies targeting the entire protein may induce a significant autoimmune response or may even inhibit normal function. To develop a safer and still effective treatment against PD, it is essential to identify functional domains which contribute to the aggregation of the human α-synuclein protein. Our group has published our results demonstrating that an α-synuclein peptide antibody targeting the N-terminal region of the protein against the peptide can help prevent neuronal loss [27]. The major pathological role of α-synuclein is attributed to the aggregated isoform, but not the monomeric form [28]. A report indicated that an antibody that targets the C-terminus of α-synuclein (9E4) could ameliorate memory/learning deficits and decrease the cortical and hippocampal α-synuclein aggregation in a human α-synuclein transgenic mouse model [29]. The accumulation and abnormal folding of α-synuclein leads to the aggregation of a toxic isoform in the body which is considered the major pathological factor of PD. In addition, using selected, specific fragments as a therapeutic target to generate antibodies against them (the detection of these fragments which can promote the aggregation of α-synuclein to form oligomers and exert their toxicity to the body) are all pivotal biomarkers for PD diagnosis or prognosis. In this regard, there is still a missing link between PD and α-synuclein. However, mounting evidence and accumulative results in AD immunotherapy research [30,31,32] have shown there is a possibility of developing effective immunotherapies against PD [33,34] if a good target is identified, though it is not a simple and easy process. So far, many research groups have tried developing immunotherapies against PD [33,34] by targeting different antigens, but minimal progress has been achieved. Educated by continuous failures from vaccines against AD [31,35], we have analyzed the antigenicity of α-synuclein, have produced several fragments, and tested them using different methods. Here we will share our results in this manuscript.

## 2. Results and Discussion

### 2.1. Epitope Mapping and Antibody Binding Domain Against Alpha Synuclein

To generate therapeutic antibodies, prediction and selection of the target antigen is the first but also the most crucial step. The full-length human α-syn protein sequence was analyzed by DNAstar software (Madison, WI, USA) using the Jameson-Wolf method to identify the high antigenic domain in its secondary structure. According to the analysis results shown in Figure 1, multiple epitopes were found which could be potential candidates for a therapeutic target. These epitopes could be recognized on the B-cell receptor found. Short peptides α-syn16–35, α-syn93-115 and α-syn116-136, were synthesized and purchased from Biomer Technology (Pleasanton, CA, USA). The recombinant protein fragments α-syn1-65, α-syn1-95, α-syn15-65, α-syn15-95 and full-length α-syn were cloned in a pGEX-6p-1 plasmid and transformed into an E. coli BL21 DE3 strain. They were then expressed and purified from the E. Coli. To generate the monoclonal antibodies (mAb) used to better recognize the spatial structure of the antigen, the full-length α-syn, and not the fragments, was used to immunize mice to obtain antibodies. To generate polyclonal antibodies (pAbs) used to better recognize the amino acid sequence, the synthetic peptides α-syn16-35, α-syn93-115 and α-syn116-136 were used to immunize goats in order to obtain antibodies as described in our previous study [27]. Goat pAbs G16-35, G93-115, G116-136 and murine mAb 1H5, 2A4, 2C6 were purified and collected. Using antibody mapping, the antigen binding sites of 2A4 and 2C6 were identified in the 93-115 region and the 116-136 region of α-syn, respectively (Table 1). While the antigen binding site 1H5 was not found, this region could be narrowed down to the aa15-65 region based on western blotting detection discussed later in this paper.

### 2.2. Recognition, Validation, and Application of Antibodies to Successfully Target Human α-Synuclein

Whether it is a polyclonal antibody or monoclonal antibody, their high affinity for human α-syn could be verified. These experimental results are displayed in Figure 2. The immunoprecipitation (IP) results further confirmed the interaction between the pAbs and the antigen. At the same time, pAbs and mAb that recognized different binding sites were also used for quantitative detection of human α-syn in human capture ELISA’s. Whether we used brain tissue or plasma, the antibodies produced in our laboratory properly recognized human α-syn. After goat pAbs were conjugated with biotin, it could also be applied to immunophenotyping blood samples. The result shown in Figure 3c is a successful example of this application. These antibodies have been useful for studying α-syn and are highly likely to be potential therapeutic antibodies.

### 2.3. Results of Aggregation and Interaction Among Fragments

A large number of present studies demonstrate the link between α-syn’s toxicity in humans to its aggregation. The co-aggregation test could confirm the interaction of full-length α-syn with α-syn residues. Since the fragments involved with the binding sites of the antibodies are used for detection in this research, the results of the assays need to be combined with several antibody probes for a comprehensive analysis. The results of the silver staining were used as a comparison of each well’s loading and aggregation. Observing the staining of the stacking gel, the upper edge of the well, and the gel of the silver staining results, the aggregation of the protein after incubation can be observed. As a result, the addition of each α-syn fragment to full-length α-syn (synthetic peptide α-syn16-35, α-syn93-115, and α-syn116-136 (Figure 4) or the recombinant proteins α-syn1-65, α-syn1-95, A-syn15-95 (Figure 5 and Figure 6) were confirmed to have the capacity to promote α-syn full-length aggregation to a certain extent.

The reason we narrowed down the potential treatment site to 15-65 is based on the following points. First, previously identified single point mutations (A30P, E46K, H50Q, G51D, A53E and A53T) in SNCA which promote the development of early onset PD were all located within this region [11,36,37,38,39,40]. There is plenty of evidence which strongly proves that the variation of these loci are highly correlated with the dysfunction and aggregation of α-syn. Mutant A30P and A53T α-syn have been demonstrated to increase fibril formation and toxicity compared with wild type α-syn [41]. The amino acid sequence alignment of α-synuclein indicated six defective repeat sequences (KTKEGV) located in the aa 1-85 N-terminal domain [42,43], and three of these are found in the aa15-65 region. Those repeats correlate with the β-sheet formation of α-synuclein. It is also indicated that this region, α-syn15-65, is crucial in the functional role of α-syn in the human body. Additionally, in the self-aggregation test of the N-terminal, it was observed that the α-syn1-65 and α-syn15-65 fragments themselves were able to be deposited without being added to full-length α-syn (Figure 7). On the other hand, in a previous publication detailing an immunotherapy trial using goat pAbs, antibody G16-35 combined with the N-terminal region and had a significant therapeutic effect in the treatment of human α-syn transgenic rats. In summary, we are confident in believing that the 15-65 region has great potential as a therapeutic target for PD patients.

### 2.4. The Cell Toxicity of Fragment α-syn 15-65

The Vero cell line and primary neuronal cells were used for cytotoxicity testing. The results showed that the fragment α-syn 15-65, although capable of forming a polymer itself or even further aggregating, did not exhibit toxicity at a concentration of 1 uM or less in the cell culture of Vero cells and primary neurons in vitro using the MTT assay (Figure 8b,e). However, if α-syn 15-65 and α-syn 1-140 were incubated for 40 h together, the toxicity of α-syn 1-140 at the same concentration could be significantly enhanced (Figure 8c,f). CellTiter-Glo luminescence assay was also used to confirm the toxicity of the α-syn 15-65 fragment. In this firefly luciferin-luciferase reaction, only ATP is readily detected by the enzymatic reaction. In the ATP test results, α-syn15-65 at 1 uM could decrease ATP generation in mouse primary neurons (Figure 8k). These MTT and CellTiter-Glo tests both indicated the negative impact of α-syn15-65 on cell viability and could further confirm that the α-syn 15-65 fragment plays a vital role in the development of PD and could be considered a candidate for a therapeutic domain (Figure 8).

### 2.5. Reaction of Antibodies to Alpha Synuclein

In the aggregation assay using antibodies with full-length α-syn, both N-terminal antibody 1H5 and C-terminal antibody 2C6 were able to reduce the formation of α-syn oligomers to a certain extent, while 2A4 had the opposite effect (Figure 9). Based on these results, mouse mAb 1H5, which recognized α-syn 15-65, has potential as a therapeutic antibody. This hypothesis may be supported by more evidence in further research.

## 3. Materials and Methods

### 3.1. Animal Study Ethics

All animals used in this research were conducted in accordance with the United States Public Health Service’s Policy on Humane Care and Use of Laboratory Animals and has been reviewed and approved by the animal component by The Institutional Animal Care and Use Committee (IACUC) at the University of South Florida.

### 3.2. Antigenicity Prediction

Full-length α-synuclein was analyzed using DNAstar software and the output was retrieved and analyzed. The performance was evaluated based on the Antigenic Index-Jameson-Wolf. Multiple potent B-cell epitope domains were provided (Figure 1).

### 3.3. Peptides Synthesis

Human α-synuclein based peptides, α-syn16-35, α-syn93-115, α-syn116-136, α-syn12-38, α-syn 1-106, α-syn 93-136, and α-syn 107-140 were synthesized and purchased from Biomer Technology (Pleasanton, CA, USA).

### 3.4. Recombinant Preparation

cDNAs corresponding to human wild type full-length α-synuclein (Hu-syn FL) and all the short fragments Hu-Syn1-65, Hu-Syn15-65, Hu-Syn1-95, and Hu-Syn15-95 were cloned in a pGEX-6p-1 plasmid and transformed into an E. coli BL21 DE3 strain. Hu-Syn FL and all other fragments were expressed and purified using Glutathione Sepharose 4B GST-tagged protein purification resin (Pittsburgh, PA, USA, GE, 17075605). Samples were lyophilized and kept at −80 °C until assayed.

### 3.5. Immunization and Hybridoma Generation

Three BALB/c mice were immunized by intraperitoneal injection with 100μg recombinant α-synuclein protein per mouse mixed with TiterMax^®^ gold adjuvant (St. Louis, MO, USA, Sigma-Aldrich, T2684). The mice were treated again at 20 and 40 days by intraperitoneal injection with 80 ug recombinant α-synuclein protein per mouse with TiterMax^®^ gold adjuvant, and then again at 70 days by intraperitoneal injection with 50 ug recombinant α-synuclein protein without an adjuvant. On day 74, two spleens were harvested, the cells were dissociated mechanically, and 3.5 × 10^8^ splenocytes were fused with 5 × 10^7^ SP2/0-AG14 mouse myeloma cells in the presence of 20 mL of 50% polyethylene glycol 1500 (St. Louis, MO, USA, Sigma-Aldrich, 10783641001). After fusion, cells were suspended in medium IMDM (Fisher Scientific, SH3022802) containing 1 X Hypoxanthine Aminopterin Thymidine (ATCC^®^ 69-X™), 15% inactivated fetal bovine serum, and 10% Hybridoma Cloning Supplement (Dallas, TX, USA, Santa Cruz Biotechnology, SC-224479) and inoculated into 12 × 96 well plates.

### 3.6. Antibody Mapping

Synthetic peptides α-syn12-38, α-syn 1-106, α-syn93-115, α-syn116-136, α-syn 93-136 and α-syn 107-140 were used to coat a 96-well plate (Immulon 4HBX) overnight at 4 °C. After removing the coating solution, the binding site was blocked using 200µl/well of 0.2% I-blocking buffer with 0.05% tween-20 and the plate was incubated for 1 h at RT, then washed four times with 1 × PBST. 50 μL of diluted mouse monoclonal antibody solution was added to the samples and were then incubated for 3 h at RT. The plate was washed four times with 1X PBST. 100 ul of secondary antibody (anti-mouse-HRP) was added and incubated at 37 °C for 45 min on a shaker, then the plate was washed again. TMB substrate was added (100 μL) and incubated for 6 min in the dark. The reaction was halted by adding 100 ul stop solution for detection at 450 nm. A 4-parameter regression was used for the standard.

### 3.7. In vitro Aggregation of α-Synuclein

Lyophilized α-syn was treated with HFIP at 10 mg/mL and then dissolved in sterile 1 × PBS for a final concentration of 200 µM. The aggregation reaction was carried out in low retention 0.7 mL centrifuge tubes. Aggregation samples for western blotting assays contained final concentrations of 25 μM of α-syn FL with or without 15 uM of α-syn fragments incubated in an orbital culture shaker and agitated at 235 rpm and 37 °C for 6 days. Aggregation samples for cell viability assays contained final concentrations of 100 μM of α-syn FL with or without 100 uM of α-syn fragments incubated in an orbital culture shaker and agitated at 235 rpm and 37 °C for 40 h. In addition to the assay just described, several aggregation publications have suggested aggregating α-syn in quiescent conditions including incubation with preformed fibrils [44] or with lipid surfaces [45]. These could be substitute aggregation protocols.

### 3.8. Immunoprecipitation

Immunoprecipitation (IP) was performed with magnetic Dynabeads M-280 Tosylactivated as described by the manufacturer (Invitrogen, Carlsbad, CA USA). Briefly, the activated 0.3 mL magnetic beads were coated with 1.5 mg goat α-syn antibody G16-35, G93-116 or G116-136. The coated beads were then harvested and resuspended in PBS with 0.1% BSA, then incubated with brain protein lysed by RIPA buffer, or plasma. The beads bound with target protein were then collected and washed three times with PBS on the magnet. The precipitates were resolved on 10% Bis-Tris gel and subjected to Western blot analysis.

### 3.9. Western Blot

Protein samples were loaded onto 10% Bis-Tris Gel and electrophoresed, then transferred onto a polyvinylidene fluoride membrane, Immobilon-P (Burlington, MA, USA, Millipore Sigma, Cat# IPVH00010, Lot# R6EA8481H). After incubating the membranes with 0.2% Iblock buffer for 1 h at room temperature, the membranes were treated with primary antibody diluted in 0.2% Iblock at 4 °C overnight. After washing and incubating with horseradish peroxidase-conjugated secondary antibodies, labeled proteins were visualized with ECL-reagent.

### 3.10. Silver Staining

The gels were stained with a silver staining kit (Waltham, MA, USA, ThermoFisher Scientific, Cat#24612) by following the manufacturer’s protocol and then developed for 5 min.

### 3.11. Primary Neuron Preparation

Primary forebrain cultures were prepared from 16–18 day embryos (12-week-old pregnant female mice). Separate cortices were carefully removed and placed in ice cold HBSS (no calcium, no magnesium, no phenol red). Cortices were dissected from brains in ice cold HBSS without Mg^2+^, Ca^2+^ and phenol red, and dissociated in ice cold HBSS without Mg^2+^ and Ca^2+^. Cultures were grown on poly-L-lysine-coated culture dishes in Neurobasal complete medium and cultured in humidified 5.0% CO2 at 37 °C for 7 days.

### 3.12. Cell Viability Assays

Cell viability was detected by MTT assay and CellTiter-Glo assay. The mouse primary neuron cells and Vero cells were seeded in 96-well tissue culture plates at 25,000 cells/well and 8000 cells/well, respectively. Primary neuron cells were cultured for 7 days, and Vero cells were cultured 1 day before treatment. The peptides were added into culture medium at various concentrations. Cells without treatment and the media background without cells were the control group and the blank group, respectively. The MTT cell proliferation kit (St. Louis, MO, USA, Sigma-Aldrich, cat 11465007001) was used to measure cell viability according to the manufacturer’s protocol. Briefly, after 24 h of treatment, 10 µL of 5 mg/mL MTT reagent was added into 100µl of culture media (final concentration 0.45 mg/mL) and incubated for 4 h at 37 °C. Then 100 µL of 10% SDS solution in 0.01 M HCl was added to each well. The culture plate stood overnight at 37 °C. Final OD values were read at 590 nm with a reference filter of 620 nm on a Bio-Rad Synergy HT plate reader. The CellTiter-Glo^TM^ luminescence assay kit (Promega Biotechnology, Madison, WI, USA, cat G7570) was used to detect ATP production as a measure of cell viability. CellTiter-Glo solution was added to samples with 100 µL of culture media as described by the manufacturer, placed on an orbital shaker for 2 min, then were incubated for 10 min to stabilize the luminescence signal which was then recorded by a plate reader. Cell viability was calculated as: Cell viability = (OD – OD blank) / (OD control – OD blank) ×100%.

### 3.13. Circular Dichroism Assay

Circular dichroism (CD)-spectra (AVIV 215 spectrometer) of peptide solutions were recorded between 195 and 260 nm using a 1 mm path-length quartz cell. Samples were diluted to 15 μM then scanned 3 times in order to find the average of the samples at each wavelength. Buffer background was subtracted.

### 3.14. ELISA Assay

Goat polyclonal antibody G93-115 (50 µL, 6µg/mL) was used as the capture antibody to coat a 96-well plate (Immuion 4HBX) for the sandwich ELISA assay. The plate was incubated overnight at 4 °C, then the coating solution was removed, and the plate was washed four times using a Bio-Tek ELX405 plate washer. The binding site was blocked with 200 µl/well of 0.2% I-blocking buffer with 0.05% tween-20 and incubated for 1 h at RT, then washed four times with 1 × PBST. Standard α-syn 93–136 solutions were prepared by serial dilution. The mouse brain tissue samples and human plasma samples were diluted to 1:14400 and 1:200, respectively, with diluent buffer containing a protease inhibitor. 50 μl samples or standard peptides were added to the wells in triplicates. 50 µL of detection antibody (G116-136) solution was added to the samples that were then incubated for 3 hrs at RT. The plate was washed four times with 1 × PBST. 100 μL of secondary antibody (anti-goat-HRP) was added and incubated at 37 °C for 45 min on a shaker, then the plate was washed again. TMB substrate was added (100 μL) and incubated for 6 min in the dark. The reaction was halted by adding 100 ul stop solution for detection at 450 nm. A 4-parameter regression method was used for the standard curve generation.

### 3.15. Immunophenotyping

Immunophenotyping analyzed by ACCURI C6 flow cytometry was performed to determine the following lymphocyte subsets: CD3+/ α-syn lymphocyte, CD19+/α-syn lymphocyte, and CD45+/ α-syn lymphocyte. Biotin was conjugated with the G116-136 antibody. 1 ml water was added into centrifuge tubes containing 50 μL human whole blood for 20 s then 110 μL 10 × PBS was added. Supernatant was removed by centrifuge at 300 g for 3 min, then repeated once more for a better result. Then 100 μL of staining buffer (0.5% FBS in PBS) and 5 μL of Goat anti-asyn116-136-biotin conjugate were added to the cell pellet and incubated at 4 °C for 15 min. After washing the cells twice with staining buffer, 10 μL of CD3-APC (BD Pharmingen, San Diego, CA, USA, Cat: 555335)/ CD19-PECy7 (eBioscience, Santa Clara, CA, Cat: 25-0198-42)/ CD45-APC(BD Pharmingen, San Diego, CA, USA, Cat: 555485) was added and then the samples were incubated at 4 °C for 15 min. Samples were read using flow cytometry after washing the cell samples with staining buffer 2 times and adding 300 μL staining buffer.

### 3.16. Data Analysis

The data manuscripts were analyzed for statistical significance using GraphPad Prism 6 software and the cell viability significant differences were analyzed with a one-way ANOVA assay using GraphPad. The level of statistical significance was set at α = 0.05.

## 4. Conclusions

Developing an immunotherapy for Parkinson’s disease has been a more realistic approach since significant achievements have been made in Alzheimer’s disease over the past 18 years [46,47,48,49]. Both active vaccine and passive immunotherapies using antibodies have showed great potential in treating AD. However, safety is still a major concern. The major breakthroughs in immunotherapy against AD is attributed to the successful identification of pathological proteins like amyloid beta (Aβ) and Tau, using them as targets for a cure [50,51,52,53,54]. Inspired by these breakthroughs in immunotherapies for AD, scientists have developed immunotherapies targeting alpha synuclein [26,54] since alpha synuclein has been identified as the main pathological protein linked to Parkinson’s Disease [55,56,57]. Though both Aβ and alpha synuclein proteins are disoriented proteins and pro-aggregation, they have their differences. Aβ builds up extracellularly and alpha synuclein builds up intracellularly [58,59]. Unlike Aβ, which is only 42 amino acids in length, alpha synuclein is bigger and contains more B-cell epitopes (see Figure 1). Therefore, alpha synuclein is more difficult to use as a therapeutic target. Several groups have identified different functional domains, such as C-terminal and N-terminal domains using different methods [60,61,62,63]. However, there still needs to be further investigation. Thus, we have synthesized three peptides and generated four recombinant truncated alpha synuclein proteins to map the functional domains of alpha synuclein. Through our collaboration, we have generated three polyclonal antibodies and tested their ability to block the pathogenesis of alpha synuclein [64]. Additionally, we tested these antibodies as a potential vaccine against PD in a PD mouse model [64]. The polyclonal antibodies and monoclonal antibodies obtained in our laboratory have a high affinity for human α-syn used in this α-syn study. The high sensitivity and good performance of these antibodies have been demonstrated in the Western blotting, ELISA assays and immunophenotyping in the results section of this publication. Thus, we have generated several mouse monoclonal anti-human alpha-synuclein antibodies and mapped them using peptides and recombinant proteins. Again, our results further demonstrated that all three major B-cell epitopes have generated correspondent antibodies against the epitopes, but the N-terminal antibody seems to be a conformational antibody because we have failed to identify the linear epitope with the peptides. To further map the function of the N-terminal, we conducted the aggregation test for α-syn, and found that the addition of all fragments used promoted aggregation of α-syn to a certain extent. The α-syn15-65 residue is capable of increasing the toxicity of accumulated α-syn in vitro and was able to aggregate without being incubated with the full-length protein. This fragment has a high correlation with disease progression which makes it a great potential therapeutic domain as a treatment site for PD. We will further validate this hypothesis in later studies. Since alpha synuclein is the major protein found in Lewy bodies, it is a pivotal finding that death is due to alpha synuclein, its fragments, the synergistic effect between full-length and fragments, or aggregation. By testing the preparations on primary neurons, we have confirmed that aggregated alpha-synuclein does have cell toxicity as opposed to monomeric alpha synuclein. We also confirmed that the 15-65 fragment can promote the aggregation of alpha synuclein aa1–140. It is worth noting that alpha synuclein, like Aβ, can bind to other proteins and mask aggregation in healthy conditions, so overexpression and reduced protein synthesis may both be related to disease onset.

The overall conclusion of this report is that we have successfully identified a potential pathological domain of alpha synuclein, and this domain may be a conformational epitope instead of a linear epitope. Our results indicate that it is possible to develop an immunotherapy by using an antigen prediction method.

## 5. Patents

Patents based on this research are pending.

## Figures and Tables

**Figure 1 ijms-20-02338-f001:**
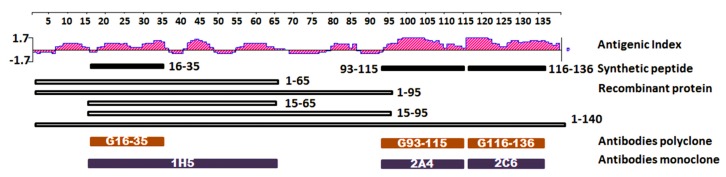
Antigenicity, recombinant protein, and related antibodies detected domains of Human α-synuclein. Antigenic Index was predicted by the Jameson-Wolf method using DNAstar. Antibodies G16-33, G93-115, and G116-136 are polyclonal antibodies produced in goats. Antibodies 1H5, 2A4, and 2C6 are all monoclonal antibodies produced in mice. Recombinant proteins were expressed in E. coli and were purified using affinity chromatography.

**Figure 2 ijms-20-02338-f002:**
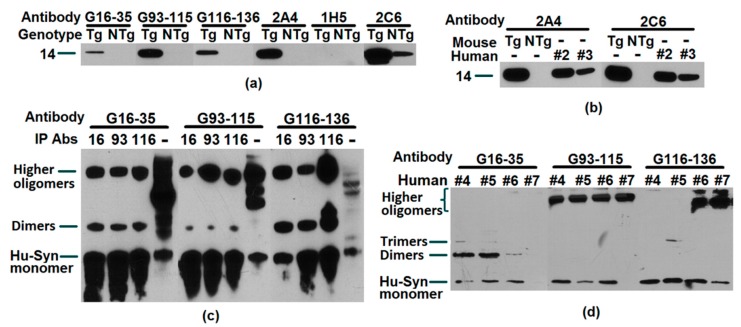
Antibodies with high affinity for human α-syn. (**a**) Western blotting result of antibodies (G16-35, G93-115, G116-136, 2A4, 1H5, and 2C6) probing α-syn in transgenic (Tg) and non-transgenic (NTg) mouse brain protein lysate; (**b**) Western blotting result of mouse monoclonal antibodies (2A4 and 2C6) probing Human α-syn in brain protein lysate of transgenic and non-transgenic mice and 2 human samples (#2 and #3); (**c**) Western blotting results of goat polyclonal antibodies (G16-35, G93-115 and G116-136) probing α-syn in human brain tissue IP with G16-33(16), G93-115(93), G116-136(116) antibodies, and human brain tissue with no-IP control; (**d**) Western blotting result of goat polyclonal antibodies (G16-35, G93-115, and G116-136) probing four human plasma samples (#4, #5, #6 and #7) IP with G93-115. The high affinity of all the antibodies listed above for human α-syn could help in future research studying human α-syn. This will be especially helpful when developing a therapeutic using an animal human α-syn model.

**Figure 3 ijms-20-02338-f003:**
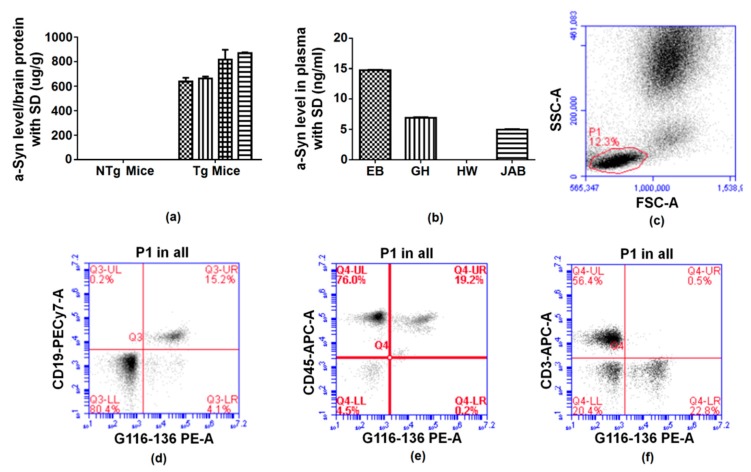
Application of antibodies. (**a**) Capture ELISA detection of human α-synuclein levels in soluble brain protein lysates of transgenic (4 Tg mice samples) and non-transgenic (4 nTg mice samples) mice by G116-136 antibody detection. (**b**) Capture ELISA result of human α-synuclein levels in 4 human plasma samples (EB, GH, HW, JAB). (**c**–**f**) Immunophenotyping result using flow cytometry to measure CD19+/α-syn, CD45+/α-syn, and CD3+/α-syn cell populations. The application of antibodies in ELISA and flow cytometry could help quantify α-syn levels in human blood and transgenic mice brain samples, as well as aid research focusing on finding potential biomarkers for PD diagnosis in the future.

**Figure 4 ijms-20-02338-f004:**
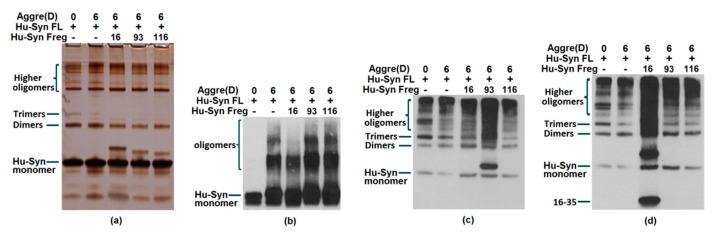
Recombinant protein full-length human α-synuclein aggregation test incubated with synthetic peptides Hu-syn16-35(16), Hu-syn93-115(93) and Hu-syn 116-136(116). Human α-synuclein aggregated with or without fragments after 6 days (6) at 37 °C using an orbital incubator or stored in −80 °C (0). Since the epitopes could be occupied by the co-incubated fragments, it is better to compare the western blot results probed with different antibodies in order to confirm conclusions made from the aggregation test. (**a**) Silver staining result; (**b**) Western blot detected by antibody 2A4; α-syn 116-136 promotes full-length α-syn aggregation; (**c**) Western blot detected by antibody G93-115; α-syn 16-35 promotes full-length α-syn aggregation; (**d**) Western blot detected by antibody G16-35, α-syn 93-115 and α-syn 116-136 promote full-length α-syn aggregation. As a result, each of these peptides α-syn 16-35, α-syn 93-115, and α-syn 116-136 could promote full-length α-syn accumulation.

**Figure 5 ijms-20-02338-f005:**
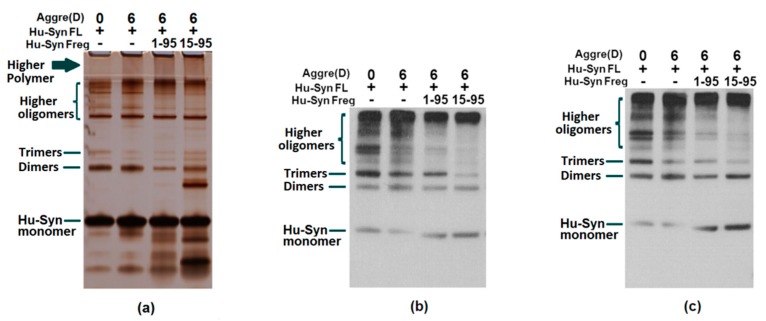
Aggregation test using full-length human α-synuclein incubated with recombinant proteins Hu-syn1-95 and Hu-syn15-95. Human α-synuclein was aggregated with or without fragments for 6 days (6) at 37 °C using an orbital incubator or stored at −80 °C (0). (**a**) Silver staining result; the silver staining results could reveal additional information within the 4% stacking gel when compared with western blotting. Polymers with larger sizes (as indicated by the blue arrow) can be seen in the silver staining which cannot be seen in the 10% Bis-Tris gel used for Western Blotting. However, fibrils may not be seen due to their large sizes using either method. (**b**) Western blot result detected by G93-115; (**c**) Western blot result detected by G16-35. Although the western results show smaller sizes (monomers, dimers, and oligomers), the silver staining result indicates that α-syn 1-95 and α-syn 15-95 actually promotes full-length protein aggregation by forming larger aggregates (indicated by blue arrow).

**Figure 6 ijms-20-02338-f006:**
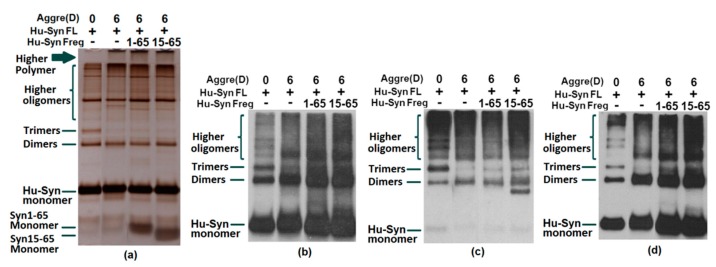
Recombinant protein full-length human α-synuclein aggregation test incubated with fragments Hu-syn1-65 and Hu-syn15-65. Human α-synuclein was aggregated with or without fragments for 6 days (6) at 37 °C using an orbital incubator or stored at −80 °C (0). (**a**) Silver staining result; (**b**) Western blot detected by antibody 2A4, (**c**) Western blot detected by antibody G16-35; d) Western blot detected by antibody 1H5. The silver staining (**a**) and western results (**b**–**d**) confirm that α-syn 1-65 and α-syn 15-65 fragments can promote full-length α-syn aggregation.

**Figure 7 ijms-20-02338-f007:**
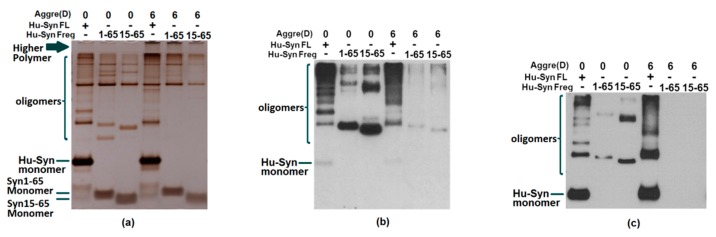
Recombinant proteins, α-syn1-65 and α-syn15-65, self-aggregation test. Full-length α-synuclein or recombinant α-syn1-65 and α-syn15-65 were aggregated for 6 days (6) at 37 °C using an orbital incubator or stored in −80 °C (0). (**a**) Silver staining result of recombinant protein aggregation; (**b**) Western blot result of recombinant protein aggregation detected by G16-35; (**c**) Western blot result of recombinant protein aggregation detected by 1H5. Residues α-syn 1-65 and α-syn 15-65 can self-aggregate.

**Figure 8 ijms-20-02338-f008:**
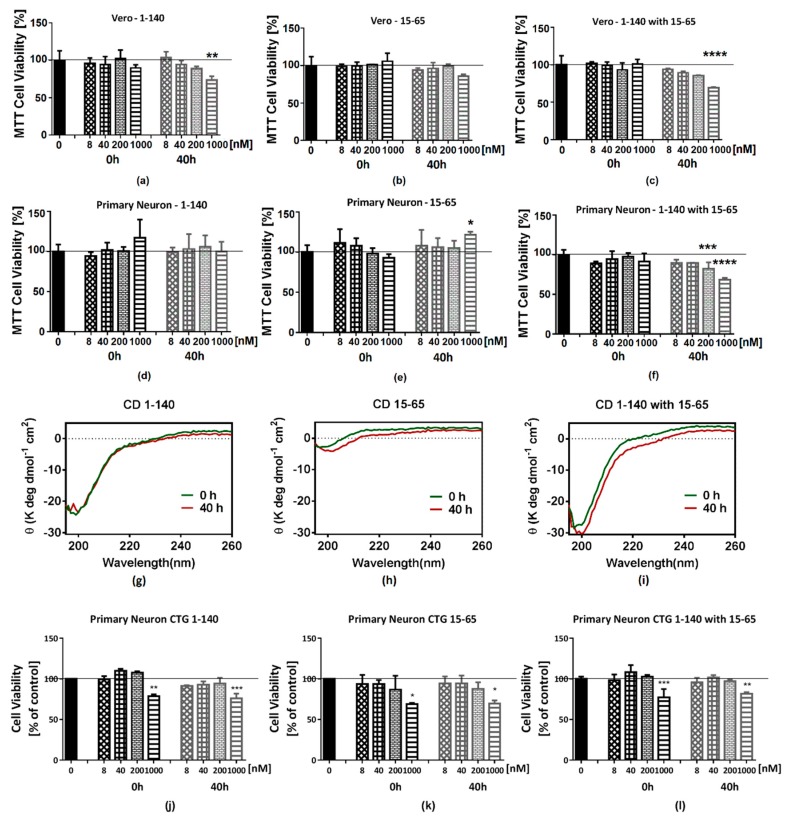
Comparative cytotoxicity of Hu-syn15-65 and full-length Hu-αsyn obtained by MTT assay and CellTiter-glo assay, after 24 h of exposure using Vero cells and primary neuron cells. Full-length Hu-αsyn, α-syn 15-65 and a co-incubation of the two were aggregated at a 100 uM concentration for 0 h or 40 h at 37 °C using an orbital incubator. (**a**–**c**) Cell viability result of vero cells using MTT assay. (**d**–**f**) Cell viability result of primary neuron cells using MTT assay. (**g**–**i**) Secondary structure changes between incubation times of the samples monitored by Circular Dichroism (CD). (**j**–**l**) Cell viability result of primary neuronal cells using CellTiter-Glo assay. In the MTT assay, α-syn 15-65 showed no toxicity in monomeric or oligomeric forms under 1000 nM after 40 h of aggregation time. However, the toxicity of the aggregated full-length α-syn increased in 1000 nM when incubated with α-syn 15-65. α-syn fragment 15-65 could increase the toxicity of the α-syn oligomer. In the CellTiter-Glo assay, it is indicated that at 1 uM of α-syn 15-65, this fragment could decrease neuron cell ATP production.

**Figure 9 ijms-20-02338-f009:**
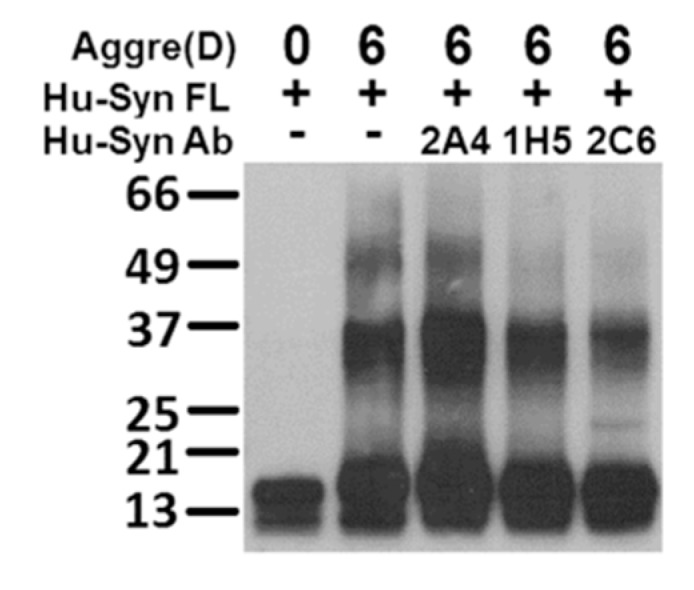
Interaction between antibodies and full-length Hu-syn obtained using Western blotting detection and probed with antibody 2A4. Monoclonal antibodies 2A4 and 2C6 could prevent the oligomeric formation of full-length Hu-syn.

**Table 1 ijms-20-02338-t001:** Mouse monoclonal antibody binding domain mapping. Antibodies were mapped using plates coated with different synthetic peptides (α-syn12-38, α-syn1-106, α-syn93-115, α-syn116-136, α-syn93-116, and α-syn107-140) for direct ELISA assays. The binding domains of 2A4 and 2C6 are human α-syn93-115 and human α-syn116-136, respectively. The binding domain for 1H5 was narrowed down to the N-terminus. Western blotting results listed below confirmed the domain located within human α-syn15-65. (- ≤ 0.05; 0.05 < * ≤ 0.1; 0.1 < ** ≤ 0.5; 0.5 < *** ≤ 1.0; **** > 1.0) In conclusion, the monoclonal antibodies that bind to the different regions of α-syn can be used to detect different fragments or synthetic peptides in this research in order to find a potential therapeutic domain. The antibody which targets the toxic domain could be a therapeutic antibody candidate.

	α-syn12-38	α-syn 1-106	α-syn93-115	α-syn116-136	α-syn 93-136	α-syn 107-140
1H5	-	*	-	-	-	-
2A4	-	-	****	-	****	-
2C6	-	**	-	****	****	****

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
