# Peer review of "Identifying the Pathological Domain of Alpha- Synuclein as a Therapeutic for Parkinson’s Disease"

_ijms, 2019, doi:10.3390/ijms20092338_

Reviewer 1 Report

The Paper by Shen et al. is an interesting study that investigates some key themes of the actual research in Neurodegenerative disease, in the specific for Parkinson’s disease (PD). The relevance for the community for new therapeutic approaches is very high given the absence of current effective treatments for PD. 

The paper is also mostly well written and the results of potential interest for the community. The identification of cross seeding of aSyn fragments with the wild type is intriguing and could be helpul to design novel therapeutic approaches. The identification of key residues in aSyn sequence is also of potential relevance for the design of therapeutics.

However, some key control experiments are still missing in the current version of the manuscript and further work is required by the authors before this manuscript can be accepted for publication.

Major Comments:

The array of antibodies, and applications suggested in figure 1-3 is interesting and well described.

The main concern of the reviewer regards the data presented in figures 4-7. The majority of the conclusions of this work regarding the fibrillation properties of aSyn and fragments identification is based mostly on the application the silver staining and western blotting methodology; while valuable, these methods by themselves are insufficient to support such complex conclusions reported by the authors.

In particular, no kinetic curve is presented in this manuscript and it is unclear if these fibrils are amyloid in nature, a key feature of the Lewy Bodies. 

In order to support their findings, the authors will need to carry out aggregation kinetics monitored by amyloid specific dyes, such as Thioflavin-T, in order to investigate how the different fragments are affecting the aSyn aggregation in vitro.

Furthermore, no further biophysical or morphological experiment is carried out to elucidate the structure of the fibrils at the end or during the reaction, thus leaving their morphology unknown to the readers. 

Methods like Atomic Force Microscopy or Transmission Electron Microscopy should be applied to obtain data on the morphological changes of these fibrils. Other biophysical methods, such as Circular Dichroism, or FTIR, should also be employed  to monitor the change in secondary structure of the different fibrils. 

It is also not clear to the reviewer what are the features of what the authors identify as "oligomers", characterisation of the biophysical features of these species  (as above) will be required.

Furthermore, the authors use a Shaking assay to induce aSyn fibrillation. While this approach has been used for many years in the community, it has now been questioned in several papers its physiological relevance. 

Several protocols have been suggested to make the aSyn aggregates in quiescente conditions, including the incubation with lipid surfaces (Galvagnion C et al, NatChemBio 2015) or preformed fibrils (Buell A et al PNAS 2014); these protocols allow to dissect the kinetic roles of Primary nucleation and Secondary nucleation in the aggregation process of aSyn; these or similar protocols will need to be applied to confirm the conclusions of the data presented in this manuscript.

Figure 8 describes the cytotoxic effects on the aSyn fragments in cell cultures using the MTT assay. While this is of interest, the MTT alone is known to be not a sufficient readout by itself for cellular health. Other readouts, such as ROS production, LDH, Calcium Influx, or Apoptosis / Necrosis assays will need to be carried out in order to effectively understand what the toxic effects of the aSyn fragments are the cell cultures.

Minor Comments:

Lines 160, 161 and 167 require bibliographic references

365-366 lines are unclear

Line 348-350: it is arguable the extent of the results obtained in terms of treating AD for immunotherapies as this disease remains uncurable to date.

Line 366: unclear sentence, please reformat for clarity

Author Response

Thank you for taking your time to review our article. I sincerely appreciate the time you have spent reviewing the article about the alpha-synuclein therapeutic domain identification. Your advice was very helpful and gave me a new perspective on our research. Following your suggestions, some updates have been made in our newest revision of the manuscript. Unfortunately, some of the data points mentioned in your review couldn’t be completed within the past 15 days. However, we will keep your comments in mind and consider them in our future studies dedicated to research in the PD field in order to contribute to a cure. 

Since the response included figures which couldn't be added in this response box, the formal response letter was attached below. 

 Again, thank you so much for your help.

Reviewer 2 Report

The authors have identified domains within alpha-synuclein, which could serve as a therapeutic target for Parkinson’s disease (PD). Novel therapeutic targets that can facilitate treatment and management of PD are urgently needed. There are elements from this manuscript that suggest that the authors may have uncovered potential therapeutic antibodies. However, some of the data presented are unclear, preliminary and inconclusive. The manuscript could have been presented and written better. The manuscript will definitely benefit from further language proofreading.

Here are some additional comments that should be addressed.

Line 86: Edit grammar in order to improve clarity

Line 90-92: Edit grammar in order to improve clarity

Line 96: “By antibody mapping” could be changed to “Using antibody mapping” ?

Line 107: “Edit grammar in order to improve clarity

Line 117: Edit grammar in order to improve clarity

Figure 2a: why is there a band for 2C6 in the non-transgenic mouse brain protein lysate? In addition, the symbols on the blot should be edited in order to improve clarity. Furthermore, the legend is not clear and hard to understand.

Figure 3: FIG 3a, what does each of the bars represent? Fig 3b, EB, GH e.tc should be defined in the legend.

Line 220: The data presented is not sufficient and clear enough to support the claims

Author Response

Response to Reviewer 1 Comments

Thank you for taking the time. I sincerely appreciate the time you spent reviewing the article about alpha-synuclein therapeutic domain identification. Your advice was very helpful and gave me a new perspective on research. We made the update following your suggestion. Any additional suggestions you may have would be welcome.

Point 1: Line 86: Edit grammar in order to improve clarity

Response 1: The original sentence “According to the analysis results shown in Figure 1, multiple candidate epitopes could be recognized by the B-cell receptor found” has been rewritten as “Figure 1, multiple epitopes were found which could be potential candidates for target. These epitopes could be recognized on the B-cell receptor found.”

Point 2: Line 90-92: Edit grammar in order to improve clarity

Response 2: “The recombinant protein fragments α-syn1-65, α-syn1-95, α-syn15-65, α-syn15-95 and full-length α-syn were obtained by expression and purification from E. coli” has been rewritten as follows:

“The recombinant protein fragments α-syn1-65, α-syn1-95, α-syn15-65, α-syn15-95 and full-length α-syn were cloned in a pGEX-6p-1 plasmid and transformed into an E. coli BL21 DE3 strain. They were then expressed and purified from the E. Coli.”

Point 3: Line 96: “By antibody mapping” could be changed to “Using antibody mapping” ?

Response 3: “By antibody mapping” changed to “Using antibody mapping”.

Point 4: Line 107: “Edit grammar in order to improve clarity

Response 4: Detailed peptides used for the mapping were added to the legend.

Point 5: Line 117: Edit grammar in order to improve clarity

Response 5: The subsection title of Figure 2.2 has been rewritten to state “2.2. Recognition, validation, and application of antibodies to successfully target human α-synuclein” and then is followed by “Whether it is a polyclonal antibody or monoclonal antibody, their high affinity for human α-syn could be verified. These experimental results are displayed in Figure 2.

Point 6: Figure 2a: why is there a band for 2C6 in the non-transgenic mouse brain protein lysate? In addition, the symbols on the blot should be edited in order to improve clarity. Furthermore, the legend is not clear and hard to understand.

Response 6: The monoclonal antibody 2C6 targets the C-terminal of alpha-synuclein. The amino acid sequence between mouse and human alpha-synuclein has no difference in the C-terminal region. Although the results of the Western Blot showed that the monoclonal antibody 2C6 is prone to binding the spatial structure of human proteins, it is still affected by the primary structure of the N-terminal. This is the reason why I only mentioned the high affinity of antibody recognition instead of the high specificity.

For the notation on the blot, I updated all the figures to improve clarity and also added more detailed information in all the legends to help improve understanding.

Point 7: Figure 3: FIG 3a, what does each of the bars represent? Fig 3b, EB, GH e.tc should be defined in the legend.

Response 7: In Figure 3-a, since each observation is an independent data set, each bar represents a different mouse. This is the same for Figure 3-b. The definitions were added to the legend to make it more readable.

Point 8: Line 220: The data presented is not sufficient and clear enough to support the claims

Response 8: The MTT results have been removed from the Figure since it did not contribute positively to this article. Instead, we used different methods and added new data sets which further support and substantiate our claims.

Round  2

Reviewer 1 Report

The paper has been improved with this round of revisions and the efforts in revising it accordingly to the reviewers comments are clear. 

- Further comments -

Point 1:

A question I still have however is if the cross seeded aggregates are amyloid in nature. as previously suggested this could be investigated by a combination of 

1) AFM / TEM to investigate the fibrillar morphology

2) Amyloid specific Dyes binding such as ThT fluorescence (upon binding of amyloid fibrils, using a fluorimeter ThT should display its characteristic shift emission maximum (from 445 nm to 482 nm), if the fibrils are amyloid-like) 

3) Secondary structure biophysical investigation i.e. using FTIR or CD. Some CD information are now already included in this new version.

Point 2: 

As the additional experiments on cellular viability are now ongoing, it will be important for the authors to include these in the final version.

I would think  these additional controls will be necessary to support the author conclusions and should be included in in a revised version of the manuscript.

Author Response

Dear Reviewer,

We really appreciated to your recommendation to our manuscripts, and we have addressed point by point.

Please find our response from the attached file.

Thank you,
Chuanhai Cao

Reviewer 2 Report

no further comment

Author Response

Thank you for taking the time to review our article.